# How Do Joint Kinematics and Kinetics Change When Walking Overground with Added Mass on the Lower Body?

**DOI:** 10.3390/s22239177

**Published:** 2022-11-25

**Authors:** Shanpu Fang, Vinayak Vijayan, Megan E. Reissman, Allison L. Kinney, Timothy Reissman

**Affiliations:** Department of Mechanical and Aerospace Engineering, University of Dayton, Dayton, OH 45469, USA

**Keywords:** biomechanics, locomotion, gait, body loading, load carriage

## Abstract

Lower-limb exoskeletons, regardless of their control strategies, have been shown to alter a user’s gait just by the exoskeleton’s own mass and inertia. The characterization of these differences in joint kinematics and kinetics under exoskeleton-like added mass is important for the design of such devices and their control strategies. In this study, 19 young, healthy participants walked overground at self-selected speeds with six added mass conditions and one zero-added-mass condition. The added mass conditions included +2/+4 lb on each shank or thigh or +8/+16 lb on the pelvis. OpenSim-derived lower-limb sagittal-plane kinematics and kinetics were evaluated statistically with both peak analysis and statistical parametric mapping (SPM). The results showed that adding smaller masses (+2/+8 lb) altered some kinematic and kinetic peaks but did not result in many changes across the regions of the gait cycle identified by SPM. In contrast, adding larger masses (+4/+16 lb) showed significant changes within both the peak and SPM analyses. In general, adding larger masses led to kinematic differences at the ankle and knee during early swing, and at the hip throughout the gait cycle, as well as kinetic differences at the ankle during stance. Future exoskeleton designs may implement these characterizations to inform exoskeleton hardware structure and cooperative control strategies.

## 1. Introduction

Studies on human walking with added mass have shown changes to the metabolic rate and, to a lesser extent, to kinematics or kinetics [1,2,3,4,5,6]. The characterization of such changes has been used to inform the designs of lower limb robotic exoskeletons and to validate predictive musculoskeletal simulations [7,8,9,10]. For example, prior research has shown that with lower extremity exoskeletons [11,12,13,14], and similarly, gait trainers [15,16], researchers will often quantify a user’s gait under three conditions: (1) without wearing the device; (2) while wearing the device but with no active control; and (3) while wearing the device and with active control. A general design guidance for such wearable robotics has been to construct it to be as lightweight as possible [17], such that the differences between the first and second conditions for metrics such as kinematics or metabolic cost can be assumed minimal. However, to validate such an assumption, such differences are quantified between the first two conditions by setting the control mode when wearing the device to be a zero-force [16], zero-torque [11,14], zero-impedance [15], or zero-assistance mode [13]. By applying such a setting, the condition decouples the active control influence from the effects of the mechanical design, primarily the added mass or inertia [16,18]. In the testing of, and comparing between, these first two conditions, many of these devices have shown such an assumption to generally be false, as they demonstrated increased metabolic cost [12,14], increased muscle activity [16], and decreased range of motion, or kinematics, oftentimes with the joint co-located to the device [11,12,16,19]. While some may argue that the third condition is intended to offset any such differences, many control schemes aim to be cooperative with the user and, as such, require information from the second condition to know how the user will respond or adapt to the perceived device [12,14,15]. In particular, control strategies such as assist-as-needed or variable impedance algorithms need to know what the user’s gait will converge to when the active control effort is off or minimized [15,20,21] in order to know how to cooperatively react. While the testing of the second condition is helpful in advancing such technology, basic science studies with added mass can also provide such insights, without being device specific. Despite the existence of prior added mass investigations (Table 1), further insight is required concerning the impact of added mass location and magnitude on gait biomechanics. For instance, spatiotemporal characteristics are generally shown to be unaffected by added mass, yet the metabolic rate has often been reported to increase. In particular, the metabolic rate has been shown to increase with +10% to +50% of body mass added at the waist [1,2,4]. Researchers have also shown metabolic rate increases with the increase in added mass magnitudes and for more distal locations [3,22]. As low as +2 kg at the foot or shank or +4 kg at the thigh had an effect [3]. At the waist, prior studies found thresholds for a significant effect to be observed at +8 kg or +6.8 kg [3,23]. Adding +5% body mass at the feet significantly increased the metabolic rate; however, no effect was observed for the same magnitude at either the shank or thigh [24].

While changes to the metabolic rate have generally agreed with respect to the factors of added mass magnitude and mass location, changes to the joint kinematics and kinetics have been less characterized. Considering the previous studies summarized in Table 1, one limitation is that only two studies thus far have tested more than 12 subjects for a given protocol [25,26], which may have limited statistical power for determining the differences between such metrics. The other limitation is that all but three studies investigated treadmill walking [1,25,27]; thus, results specific to overground walking are minimal. With that being said, a prior treadmill study, with five subjects, found no changes in the lower limb joint kinematics with up to +16 kg at the waist or thighs, or up to +8 kg at the shanks or feet, but did show increased hip joint moments with +8 kg at the feet [3]. However, another treadmill study, with 28 subjects, found that adding 3.27 kg to 9.8 kg across the pelvis and both thighs resulted in changes to the lower-limb joint kinematics and kinetics [26]. Similarly, another treadmill study, with 10 subjects, showed that +25% to +50% body mass around the waist did change the peak knee and ankle angles by up to 6% and increased the ankle moments during 30–70% of the gait cycle [5]. Finally, treadmill studies considering unilateral added mass found lower thresholds, as little as +0.3 kg at the shank, generated reduced peak knee flexion and peak ankle plantarflexion angles [6,28].

When reviewing an overground study, with 12 subjects, it has been shown that +15% body mass in a backpack resulted in increased hip and knee flexion angles at initial contact and increased joint moments with respect to ankle plantarflexion, knee extension, and hip flexion [27]. However, a further overground study, with 20 subjects, found that when adding mass to only the thighs, a subject-specific threshold existed, below which metrics such as peak knee flexion angle and knee extension moment were unchanged [25].

Thus, there still exists some ambiguity within the literature in understanding the influence that added mass may have on gait, which is particularly true with respect to lower body kinematics and kinetics. Information on lower body kinematics and kinetics not only reveals how gait is controlled, but can also help cooperative exoskeletons set their baseline trajectories. To expand on this claim, most of the prior studies listed have primarily focused on the effects of added mass by analyzing the joint kinematics and kinetics at discrete points that occur during the gait cycle [1,2,3,4,5,6,23,25,27,28], but have not quantified the impact across the whole gait cycle. Our group previously applied statistical parametric mapping (SPM) to understand such effects during treadmill walking [26,29]. This previous study, to our knowledge, was the first study to apply SPM analysis to an added mass study. The results showed that added mass caused kinematic and kinetic changes during regions of the gait cycle that would not be observed by analysis of peaks alone. A highlight of applying SPM analysis to gait biomechanics is that it can reveal the significant gait changes caused by the temporal shift of certain parts of a gait trajectory, be it joint kinematics or kinetics. Knowledge of regions of the gait cycle that are impacted by wearing added mass may be valuable to designing timing durations for assist-as-needed or cooperative control strategies of exoskeletons. Further, the comparison of the results from the analysis of peak metrics and analysis across the gait cycle may be valuable as biomechanics researchers adopt new techniques for analysis, such as SPM [30].

Thus, to add to the understanding of added mass into gait biomechanics, this study aims to characterize, where peak and SPM analyses are in agreement, the impact that factors of magnitude and location of added mass have on sagittal-plane lower-limb kinematics and kinetics during overground walking. The central hypothesis is that, with a relatively larger subject size, changes in joint biomechanics would be statistically significant across both analytical techniques and characterizable, as long as the location-specific thresholds of the added mass were reached or exceeded. The magnitudes of added mass were chosen near location-specific thresholds from the prior studies discussed and within a relevant wearable technology range, such that the total added mass would be between the lightest and heaviest devices listed within a recent review article detailing lower-extremity exoskeletons [7]. The choice of added mass was also determined by the common and commercially available products used in the field of resistance training to enable scientific reproducibility. Overall, this approach is proposed to allow for such a basic science approach to guide exoskeleton design and control, particularly those algorithms that implement states with zero active control conditions.

## 2. Methods

This study was approved by University of Dayton’s Institutional Review Board. Twenty-four healthy, young individuals without a current or a history of major lower-limb injury gave written informed consent and participated. The data from 19 participants were analyzed (10 females and 9 males, age: 22.8 ± 3.6 years, mass: 69.8 ± 12.5 kg, height: 1.760 ± 0.082 m, and BMI: 22.46 ± 3.27 kg/m^2^, all mean ± standard deviation), as shown in Appendix A. The data for five participants were omitted due to the poor quality of the recorded ground reaction force data that would have resulted in those participants not having balanced datasets for inclusion in the SPM analyses.

### 2.1. Experimental Conditions and Data Collection

Each participant performed overground walking at a self-selected speed for a Baseline condition with no added mass and for six independent added mass conditions with either a Low or High Mass Amount at one of three Mass Locations: Pelvis (+8 or +16 lb), each Thigh (+2 or +4 lb), or each Shank (+2 or +4 lb), as shown in Table 2. The condition order was randomized, and the mass was added by inserting sandbags into commercial resistance training, or weight, belts wrapped around each segment. Retroreflective markers (*n* = 43) were placed on each participant’s torso, pelvis, thighs, shanks, and feet (see Appendix A for marker and added mass location details). Before data collection for each condition, the participant walked for 2 min overground to acclimate to the added mass. For data collection, the participants walked along a 23 m walkway for 5 trials for each experimental condition, with each trial including 1 right leg gait cycle. During each trial, retroreflective marker trajectories were recorded at 150 Hz by a Vicon motion capture system with 8 cameras surrounding the walkway (Oxford Metrics, Oxford, UK). Two in-ground force plates located in the walkway (1500 Hz, Bertec, Columbus, OH, USA) recorded the ground reaction forces. The marker trajectories and ground reaction forces were converted into files for processing in OpenSim [31,32].

### 2.2. Data Processing

The ground reaction forces were processed with a Butterworth filter (4th-order lowpass, 15 Hz cutoff). The generic OpenSim Gait2392 model was scaled to match each participants’ physical dimensions per the Baseline static trial markers. Unique OpenSim mass profiles were created for each added mass condition, with matched segment mass and inertia. Sagittal-plane hip, knee, and ankle joint angles and internal joint moments were calculated with OpenSim’s Inverse Kinematics and Inverse Dynamics tools. Inverse kinematic tracking errors were minimized following guidelines from OpenSim [31,32]. The joint moment data were normalized by the participants’ body mass during the Baseline static trial.

### 2.3. Data Analysis

The joint angle and joint moment data were time normalized to the right leg gait cycle. Left-foot ground reaction forces during the first double support phase of the right leg were not available, as only two force plates were available during data collection. Therefore, the joint moment data were not analyzed during this gait phase. A custom MATLAB (MathWorks, Natick, MA, USA) code was used to extract the peak gait variables (Figure 1) and “boxplot2” [33,34], a free and open-source MATLAB package, was used to create the box plots.

For a healthy population, the gait cycle for each leg consists of a stance phase (~60% of the gait cycle) and a swing phase (~40% of the gait cycle). A stance phase of a leg starts when its heel makes initial contact with the ground (heel-strike at 0% of the gait cycle) and ends when its toes are lifted off the ground (toe-off). The swing phase of a leg begins at toe-off and ends at the next heel-strike of the leg (at 100% of the gait cycle), which then starts a new gait cycle of the leg. Over each gait cycle, kinematics (joint angle) and kinetics (internal joint moment) at the lower-limb joints, hip, knee, and ankle, are cyclic with variabilities (Figure 1). For both the kinematics and kinetics in this study, the positive and negative directions will, respectively, be as follows: hip flexion and extension; knee extension and flexion; and ankle dorsiflexion and plantarflexion. As such, the kinematic peaks analyzed in this study are as follows: (1) at the hip joint, the flexion and extension angles during the Early and Mid-to-Late Stance phases, and the flexion angle during the Swing phase; (2) at the knee joint, the flexion angles at initial contact (heel-strike, 0% of the gait cycle), during the Early Stance and the Swing phases, and the extension angle during the Mid-to-Late Stance phase; (3) at the ankle joint, the plantarflexion angles during the Early Stance and the Mid-to-Late Stance phases, and the dorsiflexion angle during the Mid-to-Late Stance phase. The kinetic peaks analyzed in this study are as follows: (1) at the hip joint, the flexion moment during the Mid-to-Late Stance phase and the extension moment during the Swing phase; (2) at the knee joint, the extension moment at the beginning of the Mid-to-Late Stance phase and the flexion moment during the Mid-to-Late stance phase; (3) at the ankle joint, the plantarflexion moment during the Mid-to-Late Stance phase. The SPM analysis included both kinematics and kinetics at the hip, knee, and ankle joints over the gait cycle. The terms and the abbreviations of the peak metrics used in this study are listed in the caption of Figure 1.

### 2.4. Statistical Analysis

For the peak analysis, the statistical significance of Mass Amount and Mass Location factors were assessed using a two-way, repeated measures ANOVA (NCSS statistical software v2021, Kaysville, UT, USA). Both Mass Amount and Mass Location were set as the within-subject factors. When the ANOVA results indicated significance (*p* < 0.05), pairwise mean comparisons were made using the Tukey-Kramer test (two-tailed). For the SPM analysis, “spm1dmatlab”, version M.0.4.8, a free and open-source MATLAB package [29,35,36] was used. A two-way, two-tailed repeated measures ANOVA and, for pairwise comparisons, two-tailed, two-sample *t*-tests (α = 0.05 for both) were conducted.

Additional metrics were analyzed and are presented as Appendix A. These include the Pearson correlations between Age/Body Mass/Body Height/BMI with respect to the peak kinematic and kinetic metrics (Appendix A), and comparisons for the range of motion (Appendix A). The probability levels and power values are also provided (Appendix A).

## 3. Results

For many of the peak analyses, both for the kinematic and kinetic metrics, and over many portions of the gait cycle in the SPM analyses, Mass Location and/or Mass Amount were statistically significant. Thus, to simplify, the results presented here are organized with respect to the lower-limb joint responses: (1) hip; (2) knee; and (3) ankle. Within each joint response, the statistically significant peak results, when compared to Baseline, are presented first (Table 3) and the SPM results are presented second (Figure 2, Figure 3 and Figure 4, Appendix A). The box plots of the peak metrics between conditions can be found in the Appendix A.

### 3.1. Hip Joint Response

#### 3.1.1. Peak Analysis

During early stance phase, Mass Amount was a significant factor for hip flexion (HF1). In particular, the High Mass amount at the Pelvis location increased the HF1 by 10.43% (3.38°, *p* < 0.001).

During the mid-to-late stance phase, Mass Amount was a significant factor for hip extension (HE). Specifically, the High Mass amount at the Pelvis location reduced the HE by 22.06% (3.08°, *p* < 0.001). Mass Amount and Mass Location were significant factors for the hip flexion moment (HF Moment). The High Mass amount at the Pelvis location increased the HF Moment by 5.74% (0.040 N·m/kg, *p* = 0.006). For added mass at the Shank location, the HF Moment was increased by 6.51% for the Low Mass amount and 12.01% for the High Mass amount (0.045 N·m/kg and 0.083 N·m/kg, both *p* < 0.001).

During the swing phase, Mass Amount was a significant factor for hip flexion (HF2). The High Mass amount at the Pelvis location increased the HF2 by 10.07% (3.32°, *p* < 0.001). Mass Amount and Mass Location were significant factors for the hip extension moment (HE Moment). The High Mass amount at the Shank location reduced the HE Moment by 10.76% (0.077 N·m/kg, *p* < 0.001).

#### 3.1.2. SPM Analysis

The SPM significance, demonstrated by the Regions of Significance (ROS) on the SPM{t} plots, showed that the High Mass amount at the Pelvis location altered the hip joint kinematics over most of the gait cycle (Figure 2), with the exception of around 30–50% of the gait cycle. The hip joint angles were reduced, resulting in greater hip flexion between 0–30% and 50–100% of the gait cycle (Figure 3). Over most of the gait cycle, no SPM significance was observed for the hip joint moments, except for briefly at around 80% of the gait cycle by the High Mass amount at the Thigh location and around 100% of the gait cycle for both the Low and High Mass amounts at the Shank location (Figure 2 and Figure 4).

### 3.2. Knee Joint Response

#### 3.2.1. Peak Analysis

During the early stance phase, Mass Amount and Mass Location were significant factors for knee flexion at both initial contact (KF1) and ~15% of the gait cycle (KF2). The High Mass amount at the Pelvis location increased the KF1 by 6.09% (0.68°, *p* = 0.026). For added mass at the Thigh location, the KF1 was reduced by 6.46% for the Low Mass amount and 10.03% for the High Mass amount (0.72° and 1.12°, *p* = 0.013 and *p* < 0.001). For added mass at the Shank location, the KF1 was increased by 8.13% for the Low Mass amount and 12.62% for the High Mass amount (0.91° and 1.41°, both *p* < 0.001). The High Mass amount at the Pelvis location increased the KF2 by 5.55% (1.33°, *p* = 0.002). The High Mass amount at the Thigh location reduced the KF2 by 5.87% (1.41°, *p* < 0.001). The High Mass amount at the Shank location increased the KF2 by 8.72% (2.09°, *p* < 0.001).

During the mid-to-late stance phase, no statistical differences were observed for the knee extension (KE) angle or the knee flexion moment (KF Moment). Mass Amount and Mass Location were significant factors for the knee extension moment (KE Moment). For added mass at the Pelvis location, the KE Moment was increased by 10.92% for the Low Mass amount and 12.64% for the High Mass amount (0.055 N·m/kg and 0.064 N·m/kg, *p* = 0.002 and *p* < 0.001). For added mass at the Shank location, the KE Moment was increased by 13.26% for the Low Mass amount and 20.92% for the High Mass amount (0.067 N·m/kg and 0.105 N·m/kg, both *p* < 0.001).

During the swing phase, Mass Amount and Mass Location were significant factors for knee flexion (KF3). At the Pelvis location, the KF3 was increased by 1.12% for the Low Mass amount and 1.27% for the High Mass amount (0.81° and 0.92°, *p* = 0.007 and *p* < 0.001). The High Mass amount at the Thigh location reduced the KF3 by 1.62% (1.17°, *p* < 0.001). For added mass at the Shank location, the KF3 was reduced by 1.10% for the Low Mass amount and 2.12% for the High Mass amount (0.79° and 1.53°, *p* = 0.009 and *p* < 0.001).

#### 3.2.2. SPM Analysis

SPM significance over small regions of the gait was observed, caused by the High Mass amount at the Thigh or Shank location (Figure 2, Figure 3 and Figure 4). The High Mass amount at the Thigh and Shank locations both increased the knee joint angles, resulting in a pose with less knee extension at around 80% of the gait cycle. The High Mass amount at the Shank location slightly decreased the knee joint moments at around 80% of the gait cycle. No sustained regions of SPM significance were observed for the Low Mass amount at any location for either knee joint kinematics or kinetics. Only a brief region of significance at around 80% of the gait cycle was observed for knee joint moment under Low Mass amount at the Shank location.

### 3.3. Ankle Joint Response

#### 3.3.1. Peak Analysis

During the early stance phase, Mass Amount and Mass Location were significant factors for ankle plantarflexion (AP1). The Low Mass amount at the Pelvis location reduced the AP1 by 20.21% (0.79°, *p* = 0.005). The High Mass amount at the Shank location reduced the AP1 by 21.05% (0.83°, *p* = 0.003).

During the mid-to-late stance phase, Mass Amount and Mass Location were significant factors for ankle dorsiflexion (AD), ankle plantarflexion (AP2), and ankle plantarflexion moment (AP Moment). For added mass at the Pelvis location, the AD was increased by 2.75% for the Low Mass amount and 5.37% for the High Mass amount (0.61° and 1.18°, *p* = 0.019 and *p* < 0.001). For added mass at the Shank location, the AD was increased by 2.70% for the Low Mass amount and 5.95% for the High Mass amount (0.60° and 1.31°, *p* = 0.023 and *p* < 0.001). The High Mass at the Pelvis location reduced the AP2 by 16.82% (1.94°, *p* < 0.001). For added mass at the Shank location, the AP2 was reduced by 14.30% for the Low Mass amount and 23.03% for the High Mass amount (1.65° and 2.65°, both *p* < 0.001). For added mass at the Pelvis location, the AP Moment was increased by 6.36% for the Low Mass amount and 9.12% for the High Mass amount (0.101 N·m/kg and 0.145 N·m/kg, both *p* < 0.001). For added mass at the Shank location, the AP Moment was increased by 2.74% for the Low Mass amount and 6.26% for the High Mass amount (0.044 N·m/kg and 0.100 N·m/kg, *p* = 0.001 and *p* < 0.001).

#### 3.3.2. SPM Analysis

The SPM analysis showed that the High Mass amount at the Thigh and Shank locations both increased the ankle joint angles at around 70% of the gait cycle, resulting in greater dorsiflexion and a less plantarflexed pose of the ankle (Figure 2 and Figure 3). The High Mass amount at each of the three locations all decreased the ankle joint moments during 40–50% of the gait cycle, resulting in greater plantarflexion moments (Figure 2 and Figure 4). The Low Mass amount at the Pelvis location resulted in the same effect on the plantarflexion moment.

## 4. Discussion

The purpose of this study was to add to the current literature, with respect to the characterization of changes to gait kinematics and kinetics when a person experiences added mass on their lower body. Within this study, every peak kinematic or kinetic metric assessed showed at least one added mass condition that was statistically significant compared to the Baseline, with the exception of the KE angle and the KF Moment. The SPM analysis showed that only a few regions of significance were identified with the Low Mass amount, while several significant regions in both kinematics and kinetics were observed with the High Mass amount. The discussion first compares the results of the SPM and peak analysis for both kinematics and kinetics. Many of the peak metrics showed large relative differences for the factors of Mass Amount and Mass Location, with some metrics only being influenced by Mass Amount, which is consistent with reports of location-specific thresholds for statistical differences [3,5,25]. As such, the discussion briefly comments on the significant outcomes of the Low Mass conditions but focuses on the significant changes occurring during High Mass conditions, as this added mass amount appeared to have exceeded many of the location-specific thresholds. Finally, responses to added mass are also discussed at the hip, knee, and ankle joint, individually.

It is noted here that in this study, the gait speeds were self-selected by the participants. With that being said, the average gait speed was 1.286 ± 0.139 m/s (mean ± standard deviation) for Baseline across all participants (Appendix A). The range of average gait speeds across all added mass conditions was 1.265 ± 0.143 to 1.292 ± 0.142 m/s (both mean ± standard deviation). Neither Thigh loading nor Shank loading significantly changed the gait speed (all *p* > 0.050). The only added mass condition that significantly changed the gait speed was the High Mass amount at the Pelvis location. This condition reduced the gait speed by 2%, relative to the Baseline (1.265 ± 0.143 m/s, *p* = 0.035). Thus, the kinematic and kinetic changes observed are believed to be in response to added mass and not a response to changes in the gait speed.

### 4.1. Comparison of SPM and Peak Analysis

#### 4.1.1. Kinematic Outcomes

The SPM analysis did not result in any kinematic differences for Low Mass conditions at any Mass Location (Figure 2). However, the peak analysis suggested several significant differences between the Baseline and Low Mass conditions. The peak behaviors appeared to be most sensitive to the Low Mass amount at the Shank location, with differences found for knee flexion just after and prior to heel strike (KF1 and KF3), and for ankle angles just after and prior to toe-off (AP2 and AD).

The largest regions of kinematic changes to the gait cycle identified by the SPM analysis were for the High Mass amount at the Pelvis (between 0–30% and 50–100%). The 0–30% region is associated with the peak analysis for hip flexion (HF1). The 50–100% region begins with the hip extension peak (HE) and ends with the pre-strike hip flexion peak (HF2). The peak analysis revealed significant changes at HF1, HE, and HF2 with the High Mass amount at the Pelvis. Specifically, the peak analysis identified that increasing the Pelvis mass produced significantly more flexed hip behavior, with increased flexion just after and just prior to heel strike (HF1 and HF2). At maximum hip extension (HE), this behavior manifested as reduced extension. The lack of SPM differences in the kinematic hip profiles, due to Thigh or Shank loading, was consistent with the peak analysis.

For the High Mass amount at the Thigh location, the characteristic region of kinematic change identified by the SPM analysis was on the knee and ankle angles between ~70–80% of the gait cycle (Figure 3). Comparing this to Figure 1 shows that this region does not correspond with any of the peak analysis locations, coming after peak knee flexion (KF3) and peak ankle plantarflexion (AP2). In the peak analysis at KF3, there is a decrease in knee flexion for adding High Mass at the Thigh location, however the peak analysis at AP2 shows no impact. This highlights that the peak analysis results might not be reflected in the SPM analysis (as with KF3) and, alternately, that the SPM could identify significant differences outside of the isolated peak locations.

For the High Mass amount at the Shank location, the characteristic region of kinematic change identified by the SPM analysis was between 70–85% for knee angles and between 60–75% for ankle angles (Figure 3). The start of the knee region corresponds with peak knee flexion (KF3). At KF3, the peak analysis showed that increased shank loading resulted in decreased knee flexion, in the same way as increased thigh loading (Figure 3). The start of the ankle region corresponds with peak ankle plantarflexion (AP2). At AP2, the peak analysis showed that increased shank loading resulted in decreased plantarflexion, which was not seen for increased thigh loading (SPM or peak analysis). This highlights that the SPM results can help identify which peak analysis points may have significant differences based on which peak events are contained in a significant SPM region.

In summary, when a significant region of difference was identified by the SPM, then any peak events contained in that region also had significant differences in the peak analysis. For the High Mass amount at the Pelvis location, this was HF1, HE, and HF2. For the High Mass amount at the Shank location, this was KF3 and AP2. The benefit of SPM analysis over peak analysis is that it provides a clearer understanding of how peak differences extend their impact across the continuous gait cycle. In contrast, several of the peak analyses identified significant differences in regions of the gait cycle where SPM did not report a difference. This inconsistency is most likely due to the variable timings of the occurrence of the peak events. The more the timing of a peak event varies, the more this peak will be diminished when many gait cycles are normalized and averaged. Several methods to address this issue have been proposed, including Dynamic Time Warping and Piecewise Event Alignment [30,37,38,39]. While the application of such methods is outside the scope of the current paper, our results suggest that such improvements to SPM approaches or a combination of SPM and peak analysis will yield the most complete understanding of the impact of an intervention such as adding mass.

#### 4.1.2. Kinetic Outcomes

When considering the SPM outcomes, the joint moment or kinetic results highlight a few considerations. First, that significant SPM regions lasting only 1–2% of the gait cycle may be present (as for shank loading impact on the hip and knee moments) but do not yield useful insights. Additionally, significant SPM results for joint moments must be examined relative to their magnitude profiles as many of the regions deemed significantly different occur at near zero magnitudes. Considering just the significant SPM regions that lasted for more than 2% of the gait cycle and occurred at non-zero magnitudes, only the ankle joint moment had significant outcomes, and all occurred in the region of the Ankle Plantarflexion Moment (AP Moment) peak. The peak analysis of the AP Moment was consistent with the SPM results. The peak analysis also resulted in significant changes for other joint moments that were not suggested by the SPM results. For example, the High Mass amount at the Shank location resulted in significant Hip Flexion (HF) Moment, Hip Extension (HE) Moment, and Knee Extension (KE) Moment changes compared to the Baseline. Overall, this was consistent with the kinematic results. Peak events were significantly different when they fell in a SPM region, but the peak analysis highlighted further differences not seen in the SPM.

### 4.2. Hip Joint Response

For the High Mass amount at the Pelvis location, in both early stance and late swing, an increase of approximately 10% in HF1 and HF2 were observed and increased hip flexion was observed by the SPM analysis (between 0–30% and 50–100%), revealing that the foot was further in front of the pelvis than normal, which is consistent with similar added mass loading [27]. As this adaptation is present in the swing phase, despite no added mass being present on the swing leg, this appears to represent a preparatory adaptation to extend the heel strike limb moment and improve the ability of the leg to manage the velocity of the pelvis during the swing-to-stance transition. The function of gait is to move the body in space, typically maintaining near constant velocity during steady-state gait, while managing step-to-step transitions with minimal energy loss. Prior work suggests that human gait achieves a low cost of transport by maximizing the passive dynamic motion of the limbs, while maintaining control of limb accelerations at toe off and decelerations at heel strike [40,41]. Pelvis loading must still be managed through adaptations in the lower limb kinematics and kinetics. However, pelvic kinematic energy is more constant than for the thigh/shank/foot, and pelvic rotation is also comparatively minimal. High Pelvis loading resulted in an approximately 6% increase in the peak HF Moment, although no SPM significance was observed. High Shank loading resulted in changes similar to pelvis loading, an approximately 12% increase in the HF Moment, with an 11% decrease in the HE Moment at terminal swing. For High Pelvis loading, the HE was decreased by 22%, which was consistent with the SPM results in that region. In contrast, the SPM analysis showed an increased hip flexion in the early stance and late swing, and the peak analysis showed that HF1 and HF2 were increased by 10%. This HE reduction places the thigh more underneath, rather than behind, the trunk. This may reflect a desire to maintain consistent step length while accommodating the contralateral limb HF increase. However, a reduction in HE is likely to reduce the trailing limb angle, the angle between a vector from the foot contact point and the center-of-mass versus vertical. When considering this HE reduction, note that for the impaired populations that might utilize an exoskeleton device, HE is often already reduced and it is an increasingly common therapeutic target [42].

### 4.3. Knee Joint Response

High Pelvis loading resulted in an increase in the peak knee joint moment, a 13% increase in KE Moment, but no significant regions were identified by the SPM analysis. High Shank loading resulted in changes similar to High Pelvis loading. Specifically, there was a 21% increase in the peak KE Moment and increased knee extension was observed by the SPM prior to the peak KE Moment. The result for the KE Moment is likely a result of increased activations in the Rectus Femoris and Vastus Medialis [43]. A limited number of statistical changes in knee kinematics and kinetics were found due to Low or High Thigh loading. Nevertheless, the kinematics contradict, while the kinetics further support a prior study [25]. Such changes include reduced knee flexion, or a more extended knee joint, by roughly 10% at KF1, or heel strike, and 5.9% at KF2, or peak weight acceptance. In contrast, shank loading resulted in increased knee flexion by around 8.1–12.6% at KF1 and 8.77% at KF2. At the moment of heel strike, the thigh is beginning to accelerate posteriorly while the shank is typically rotating and moving anteriorly, right up until heel strike. Added mass at the thigh during this phase can move more passively with gravity, while added mass at the shank must be rotated and lifted against gravity to achieve a suitable heel strike position. Thus, for shank loading, the body appears to adopt a slightly more flexed position at heel strike that is sustained throughout early stance. As crouched gait patterns are a common pathology and known to be energetically costly [40], the interplay of thigh and shank loading on knee flexion/extension should be further studied.

### 4.4. Ankle Joint Response

Low Pelvis loading results in a decrease of approximately 20% in AP1, and High Shank loading results in an approximately 21% decrease in AP1. Similar reductions in ankle plantarflexion after AP1 for High Shank loading were observed in the SPM. Prior work interprets such ankle plantarflexion change during early stance as an energy absorption or weight acceptance response [41]. High Pelvis loading resulted in a consistent increase in joint moments across the hip, knee, and ankle, which were apportioned relatively equally. Specifically, in addition to the approximately 6% increase in the HF Moment and the 13% increase in the KE Moment, a 9% increase in the peak AP Moment and a sustained region of SPM significance in the ankle plantarflexion moment was observed at the ankle joint. These moments characteristically peak during late stance and are viewed as generating propulsive impact on the body [44]. Such increases have been previously observed for higher loadings and are supported by the increased Gastrocnemius activation found by others and in our parallel analysis [5,43]. High Shank loading resulted in changes similar to pelvis loading. Specifically, in addition to the approximately 12% increase in the HF Moment and the 21% increase in the KE Moment, a 6% increase in the AP Moment and a sustained region of SPM significance in the ankle plantarflexion moment was observed at the ankle joint. Beyond this, shank loading also resulted in a 6% increase in AD and a 23% decrease in AP2. Overall, this is consistent with the idea that late stance moments manage the center of mass propulsion and limb transition to swing [40,41]. A primary difference was that the moment increases for shank loading were roughly twice the percentage increase versus the pelvis and with only half the mass amount. Thus, while sensitivity is higher for added mass at the shank, the response characteristics are similar. The increased ankle moment and increased trailing limb angle in late stance have both been shown to support increased propulsion [45]. In combination with the HE reduction, AD was increased by 5–6% and the AP Moment was increased by 9%, which may have been due both to the impact of loading and the subsequent kinematic changes. In a healthy gait, the peak ankle moment production occurs at about 10° ankle dorsiflexion and reduces as the ankle moves toward plantarflexion [44]. Interestingly, while pelvis or shank loading generated similar responses during the mid-to-late stance, the HE reduction was only present for pelvis loading. This may be due to the fact that decreased HE would likely increase the moment arm between shank added mass and foot center of pressure, requiring additional ankle moment increases if this position was adopted.

### 4.5. Limitations

Due to time constraints, only five gait cycles were captured for each added mass condition for each participant. Due to limited space on the waist, retroreflective markers were placed over pelvis landmarks but on the waist belt which might reduce the accuracy of pelvis motion and hip joint location. The resistance training, or weight, belts used to secure added mass to each segment remained on throughout the study, even when no mass was added to that segment. This may have had some minimal impact on the participant but allows consistency in comparing the added mass impact across all conditions.

## 5. Conclusions

In summary, the Low Mass amount at any of the three locations (pelvis, thigh, or shank) has been shown to exceed a threshold for statistical changes of at least one gait peak kinematic and/or kinetic metric but resulted in only increased ankle plantarflexion moment with Pelvis loading identified by SPM. The High Mass amount at any location has been shown to yield multiple kinematic and kinetic changes, which are both significant for the peak and SPM analyses. Thus, the study’s original hypothesis is supported at the High Mass amount, as it is not only significant but also characterizable. At the hip joint, the High Mass amount at the Pelvis increased the hip flexion angle in early stance and swing and increased the hip extension in late stance. The High Mass amount at the Shank increased knee flexion angle during swing. Finally, at the ankle, the High Mass amount at the Shank increased ankle plantarflexion in late stance and the High Mass amount at the Pelvis, Thigh, and Shank locations increased the ankle plantarflexion moment during stance.

All of the peak kinematic and kinetic events achieved significance when they fell in a region of significance in the Statistical Parametric Mapping (SPM) analysis. However, peak analysis highlighted further differences not seen in the SPM, supporting the idea that SPM may require multiple tests or additional analyses to determine changes due to timing and magnitudes of the biomechanical data [30]. This study adds knowledge to healthy adults’ kinematic and kinetic responses to adding mass on the lower limbs during overground walking. This knowledge details and characterizes the wearers’ adaptations to the additional mass by an exoskeleton, which can help inform the designs of the control algorithms and the structure of future wearable assistive devices.

To conclude, the agreement between the peak analysis and the SPM analysis suggests that certain biomechanics differences would be expected in healthy adults in response to the mass of an exoskeleton. For example, from this study, it is now understood that healthy adults will increase the AP Moment for the High Mass amount at any of the Pelvis, Thigh, or Shank locations, and that the increases start occurring at ~10% of the gait cycle before the peaks and end at the peaks. Additionally, with a relatively higher mass amount, the High Mass amount at the Pelvis location, in this case, the differences extend beyond the peak value, spanning across approximately 20% of the gait cycle around the peaks. This suggests that an exoskeleton may need to not only provide assistance dependent on the amount of the loading, but also vary the level of assistance at different portions of the gait cycle. On the other hand, although added mass showed significance for many peak metrics, its impact on the hip and knee joints during the stance phase never extends beyond the peaks. This suggests that the adaptations to exoskeleton mass may be relatively small over most of the stance phase at the hip and knee joints, and the adaptations may be concentrated at the peaks. To distribute theses adaptations across the gait cycle, an exoskeleton may intercept earlier as it approaches the peaks. These observations apply to exoskeletons with a similar mass distribution to those tested in the current study and to those with low joint stiffness and damping.

## Figures and Tables

**Figure 1 sensors-22-09177-f001:**
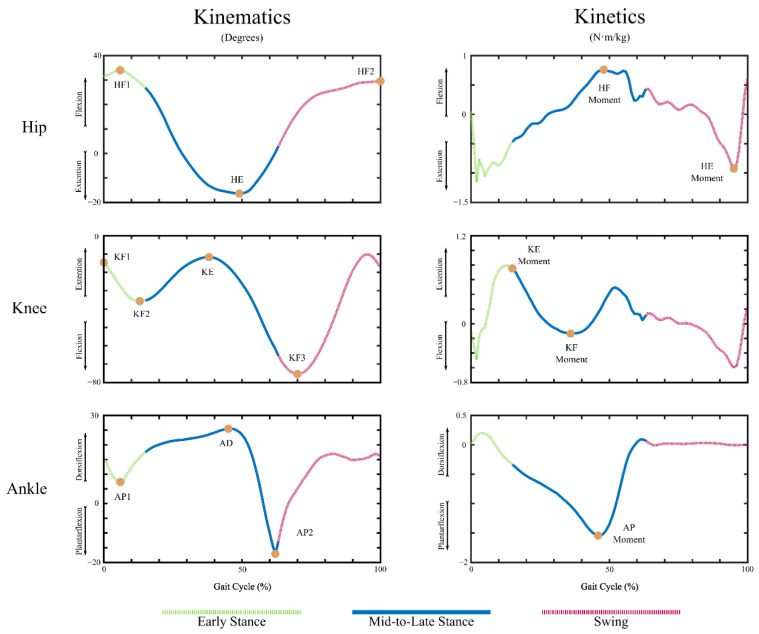
Kinematic and kinetic peak metrics marked on top of gait kinematics and kinetics of a representative trial. Gait kinematics and kinetics have been color-coded to reflect Early Stance (green), Mid-to-Late Stance (blue), and Swing (red) phases. Full metric names and their abbreviations are as follows: Stance Maximum Hip Flexion (HF1), Stance Maximum Hip Extension (HE), Swing Maximum Hip Flexion (HF2), Maximum Knee Flexion during Initial Contact (KF1), Stance Maximum Knee Flexion (KF2), Mid to Late Stance Maximum Knee Extension (KE), Swing Maximum Knee Flexion (KF3), Stance Maximum Ankle Plantarflexion (AP1), Maximum Ankle Dorsiflexion (AD), Maximum Ankle Plantarflexion (AP2), Stance Maximum Hip Flexion Moment (HF Moment), Swing Maximum Hip Extension Moment (HE Moment), Stance Maximum Knee Extension Moment (KE Moment), Mid-to-Late Stance Maximum Knee Flexion Moment (KF Moment), and Maximum Ankle Plantarflexion Moment (AP Moment).

**Figure 2 sensors-22-09177-f002:**
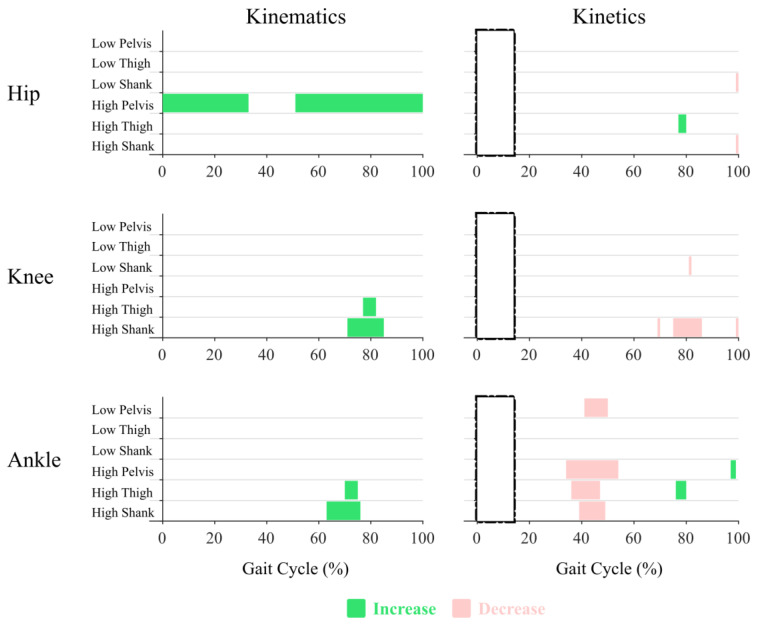
Joint kinematics and kinetics SPM significance highlighted in color over the gait cycle. Significance is achieved when SPM{t} exceeds its critical value. Comparisons are all with respect to the Baseline condition. Rectangles with dash-dotted edges are used to mask the early stance joint kinetics for lack of necessary ground reaction force data.

**Figure 3 sensors-22-09177-f003:**
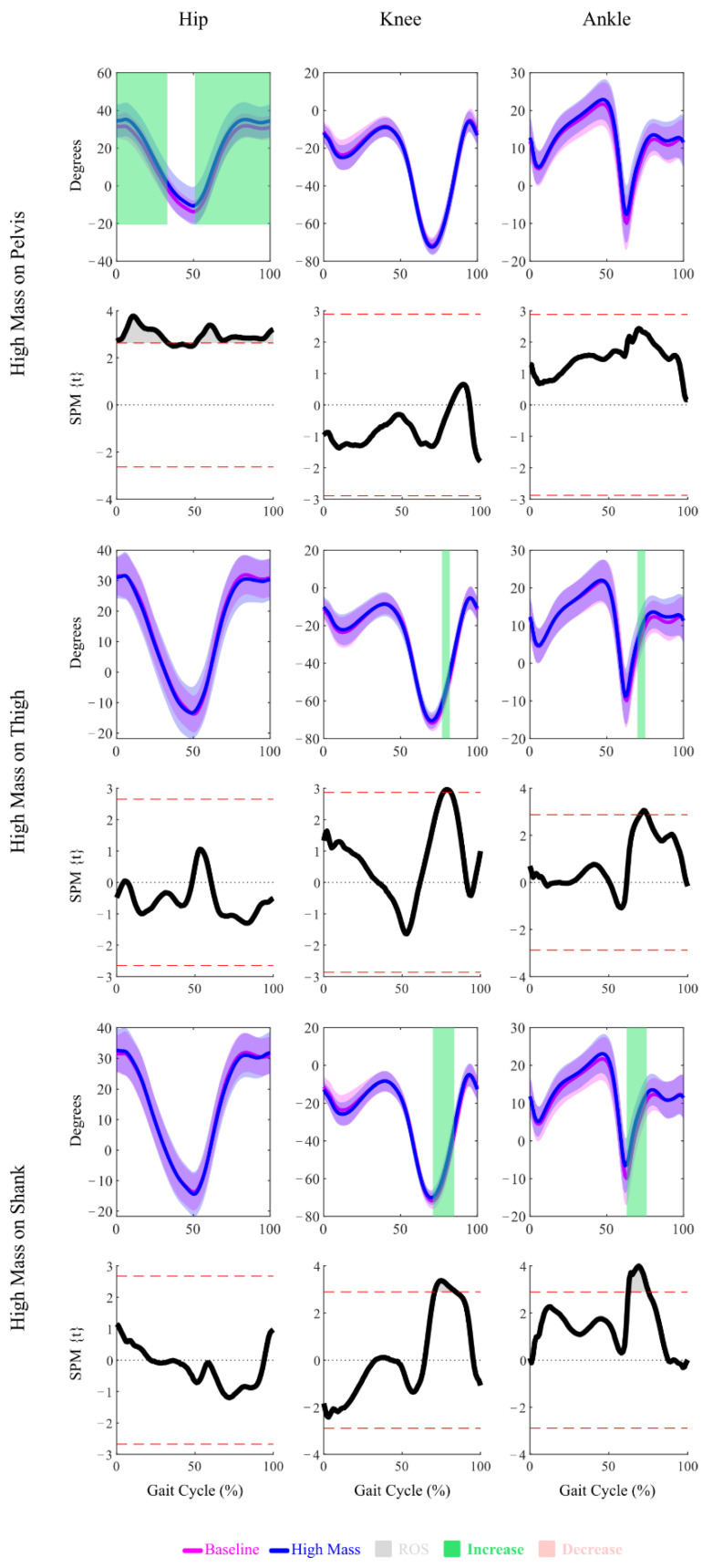
Mean and Standard Deviation plots of the joint kinematics under both the High Mass conditions and the Baseline condition, and SPM{t} plots demonstrating the significance (Region of Significance, or ROS) of the comparisons between the High Mass and the Baseline conditions. Upon significance, directions of changes are highlighted in color.

**Figure 4 sensors-22-09177-f004:**
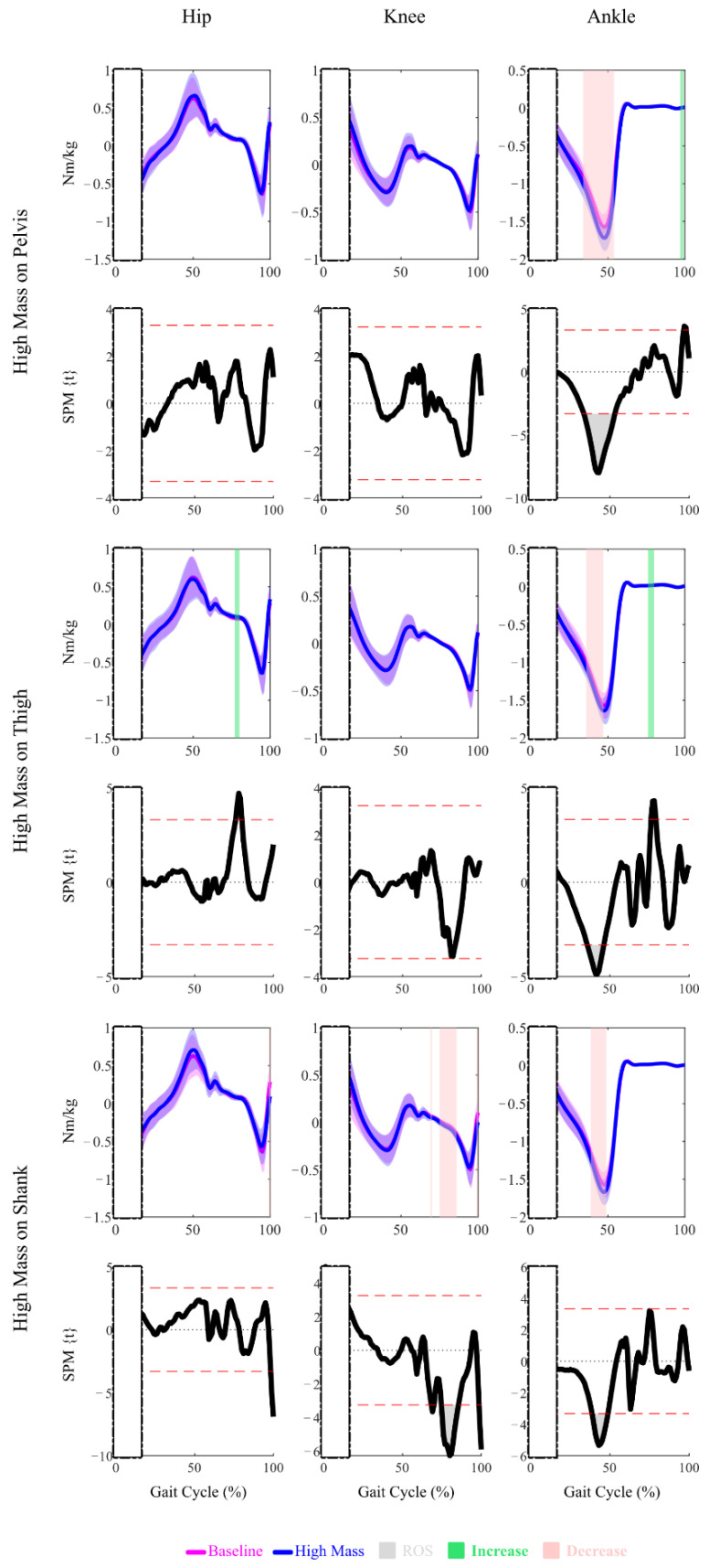
Mean and Standard Deviation plots of the joint kinetics under both the High Mass conditions and the Baseline condition, and SPM{t} plots demonstrating the significance (Region of Significance, or ROS) of the comparisons between the High Mass and the Baseline conditions. Upon significance, directions of changes are highlighted in color. Rectangles with dash-dotted edges are used to mask the early stance joint kinetics for lack of necessary ground reaction force data.

**Table 1 sensors-22-09177-t001:** Selection of previous studies featuring kinematic/kinematic/metabolic cost analysis during human locomotion with added mass or inertia.

Ref.	Subjects	Age (Years)	Body Mass (kg)	Motion	Mass		Changes Observed		
					Amount	Location	Spatiotemporal	Metabolic Rate	Kinematic/Kinetics
[1]	8	26 ± 5	68.7 ± 12.5	TW OW ^1^	0–0.3 BM	Hip	✓	✓	
[2]	10	NA	68.65 ± 8.1	TW	0–0.5 BM	Hip		✓	
[3]	5	NA	74.16 ± 5.18	TW	0–16 kg	Various	✓	✓	✓
[4]	10	32 ± 7	63.3 ± 9.8	TR	0.1–0.3 BM	Hip		✓	
[5]	10	21–45	67.1 ± 8.5	TW	0.25–0.5 BM	Trunk			✓
[6]	9	25.1 ± 5.2 ^2^ 30.9 ± 10.3 ^3^	72.4 ± 12.5 ^2^ 74.9 ± 9.0 ^3^	TW	0.7–10.2/0.6–5.2 kg ^2^ 0.3–3.5 kg ^3^	Various			✓ ^3^
[22]	8	26.8 ± 2.7	74.9 ± 9.2	TW	0.5–2 & 2–22.1 kg	Various		✓	
[23]	8	19–26	71.1 ± 12.0	TW	15–45 lb	Trunk	✓	✓	
[24]	12	22.9 ± 2.2	70.6 ± 13.7	TW	0.05 BM	Various		✓	
[25]	20	22.7 ± 1.8	66.2 ± 5.7	OW	Subject-specific	Thigh			
[26]	28 ^4^	22 ± 4 48 ± 9	71.0 ± 10.5 76.5 ± 13.0	TW	3.27–9.8 kg	Various			✓
[27]	12	23 ± 3	70.9 ± 12.7	OW	0.15 BM	Trunk	✓		✓
[28]	12	22–31	71.1 ± 8.0	TW	2.7 & 0–3.6 kg	Various	✓		✓

OW: Overground Walking, TW: Treadmill Walking, and TR: Treadmill Running. BM: Body mass. ^1^ Treadmill walking for metabolic rate analysis and overground walking for spatiotemporal analysis. ^2^ Ankle loading. ^3^ Pelvis loading. ^4^ 14 young adults and 14 middle-aged adults.

**Table 2 sensors-22-09177-t002:** The seven experimental conditions included a Baseline condition with no added mass and six added mass conditions varying Mass Amount and Mass Location.

Mass Amount	Mass Location	Total Added Mass (lb)
Baseline		+0
Low	Pelvis	+8
Low	Thigh ^1^	+4
Low	Shank ^1^	+4
High	Pelvis	+16
High	Thigh ^1^	+8
High	Shank ^1^	+8

^1^ Mass was added to the location on both legs.

**Table 3 sensors-22-09177-t003:** Mean and Standard Deviation values of peak kinematic and kinetic metrics.

	**Added Mass**		**Kinematics (Degrees)**				**Kinetics (N·m/kg)**	
	Amount	Location	*HF1*	*HE*	*HF2*		*HF Moment*	*HE Moment*
			Flexion (+)	Extension (−)	Flexion (+)		Flexion (+)	Extension (−)
Hip	Baseline		32.366 (6.761)	−13.976 (6.023)	32.911 (6.400)		0.690 (0.226)	−0.713 (0.218)
	Low	Pelvis	33.351 (7.637)	−13.866 (7.585)	33.778 (7.015)		0.722 (0.263)	−0.731 (0.229)
	Low	Thigh	32.425 (7.342)	−13.903 (7.832)	32.816 (7.314)		0.705 (0.233)	−0.746 (0.226)
	Low	Shank	32.856 (7.675)	−14.641 (7.708)	32.763 (8.111)		**0.735 (0.241) ^1^**	−0.682 (0.232)
	High	Pelvis	**35.743 (8.869) ^2^**	**−10.893 (10.083) ^4^**	**36.227 (8.948) ^2^**		**0.730 (0.235) ^1^**	−0.743 (0.218)
	High	Thigh	32.226 (7.467)	−13.580 (8.463)	31.812 (7.574)		0.673 (0.227)	−0.717 (0.216)
	High	Shank	33.159 (7.286)	−14.630 (7.284)	32.955 (6.984)		**0.773 (0.224) ^2^**	**−0.636 (0.213) ^2^**
	**Added Mass**		**Kinematics (Degrees)**				**Kinetics (N·m/kg)**	
	Amount	Location	*KF1*	*KF2*	*KE*	*KF3*	*KE Moment*	*KF Moment*
			Flexion (−)	Flexion (−)	Extension (+)	Flexion (−)	Extension (+)	Flexion (−)
Knee	Baseline		−11.197 (5.825)	−24.001 (8.071)	−8.071 (5.564)	−72.122 (4.296)	0.503 (0.244)	−0.297 (0.150)
	Low	Pelvis	−11.647 (5.431)	−24.980 (6.839)	−8.002 (5.563)	**−72.929 (4.739)**	**0.558 (0.223) ^2^**	−0.319 (0.230)
	Low	Thigh	**−10.474 (5.654) ^1^**	−23.828 (7.439)	−7.883 (5.962)	−72.034 (5.083)	0.531 (0.231)	−0.300 (0.146)
	Low	Shank	**−12.108 (5.744) ^1^**	−24.828 (7.685)	−7.383 (5.370)	**−71.328 (4.381)**	**0.570 (0.220) ^2^**	−0.311 (0.149)
	High	Pelvis	**−11.879 (5.258) ^1^**	**−25.334 (6.562) ^1^**	−8.608 (5.233)	**−73.038 (4.094)**	**0.567 (0.246) ^2^**	−0.314 (0.159)
	High	Thigh	**−10.074 (5.833) ^2^**	**−22.593 (8.032) ^1^**	−8.323 (6.498)	**−70.954 (4.434)**	0.482 (0.225)	−0.306 (0.171)
	High	Shank	**−12.611 (4.726) ^2^**	**−26.095 (6.234) ^1^**	−8.090 (5.244)	**−70.596 (3.843)**	**0.608 (0.229) ^4^**	−0.313 (0.162)
	**Added Mass**		**Kinematics (Degrees)**				**Kinetics (N·m/kg)**	
	Amount	Location	*AP1*	*AD*	*AP2*		*AP Moment*	
			Plantarflexion (−)	Dorsiflexion (+)	Plantarflexion (−)		Plantarflexion (−)	
Ankle	Baseline		3.923 (4.691)	22.048 (5.585)	−11.509 (7.025)		−1.594 (0.173)	
	Low	Pelvis	**4.716 (3.955) ^4^**	**22.654 (5.047)**	−10.969 (6.950)		**−1.696 (0.222) ^1^**	
	Low	Thigh	4.414 (3.905)	22.584 (5.471)	−10.597 (7.379)		**−1.627 (0.163)**	
	Low	Shank	4.412 (4.149)	**22.644 (5.373)**	**−9.864 (8.075) ^2^**		**−1.638 (0.163)**	
	High	Pelvis	4.398 (4.572)	**23.232 (5.132) ^1^**	**−9.574 (7.376) ^3^**		**−1.740 (0.173) ^1^**	
	High	Thigh	4.169 (4.568)	22.305 (5.372)	−10.700 (7.275)		**−1.664 (0.182)**	
	High	Shank	**4.749 (4.053) ^4^**	**23.360 (5.159) ^1^**	**−8.859 (6.982) ^4^**		**−1.694 (0.172) ^1^**	

Bold items are significantly different from the baseline condition during the pairwise comparisons (*p* < 0.05). Differences from the Baseline condition: ^1^ ±5–10%; ^2^ ±10–15%; ^3^ ±15–20%; ^4^ >±20%. Abbreviations for the peak metrics are defined in Figure 1.

## Data Availability

Not applicable.

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
