# Peer review of "How Do Joint Kinematics and Kinetics Change When Walking Overground with Added Mass on the Lower Body?"

_sensors, 2022, doi:10.3390/s22239177_

Round 1

Reviewer 1 Report

This is a very well written paper, but I would suggest some changes/additions:

1. I have rarely if at all read a paper with a question in the title. I think it would sound better if it is called "Change of Joint Kinematics and Kinetics when..."

2. For benefit of this journal's general reader, please add general, detailed information on gait kinetics and kinematics. You could place it in the introduction or in a separate section.

3. Please explain your specific choice of repeated measures ANOVA and Turkey-Kramer in section 2.4.

Reviewer 2 Report

I find the research very interesting and novel for this journal. However, it is required to attend the following comments:

·         The Introduction section can be more concrete. For example, the authors should consider highlighting the problem they are trying to solve in the proposed experiments.

·         In addition, the Introduction section should also detail the open problem they want to resolve.

·         The main area of improvement for the current presentation is to demonstrate the novelty and significance of the proposed experiments in the methodology.

·         It is required to explain the motion capture equipment used to monitor the walking cycles of the volunteers, as well as the configuration of the scenario proposed.

·         The conditions in which it was carried out would be helpful. For example, to have a flow diagram of the scenario where the study was constituted.

·      Many terms are not clearly defined formally.

·         The experiment section is hard to follow. It is unclear about the experiment setup and how to interpret the results in the tables. The authors should consider adding figures to provide more insights about the experiments.

In the section on the conclusions and discussions, it is mentioned that some motion capture equipment was used which, at the time of placing the markers on the reference point of the pelvis, on the belt, the authors dictate that the results present a significant error, this particular situation is not clear in the work what impact it will generate on the results,

If it is the case that it was taken into account, how does this affect the results?

How were adjust the results when this error was present?

If it was not considered, how reliable would the results be?

Reviewer 3 Report

Although the study topic is not novel and some similar studies have been published before, there are a little more participants involved in this study, therefore the statistical power could be increased in this study. In addition, the present manuscript is well-written and presented in great detail. Some suggestions for the authors to further improve the manuscript are provided as below.

* The reviewer has some suggestions for Figure 1.

First, there is a problem that the light blue color in the three sub-figures in the first column is different from that in the second column. The same color should be used for the same gait phase.

Second, the reviewer suggests to replace light blue with green or other colors except for blue, or it could be a little bit difficult for the readers to distinguish the light blue from the deep blue.

Third, please clearly state in the figure legend that which color stands for which gait phase.

* Lines 137-139: 24 participants were recruited, but only the data of 19 of them were analyzed. The authors must clearly state why 5 of them were omitted. It is not allowable to vaguely state that the data were omitted “due to technical issues”.

* Line 143: Speed could be an important factor influencing the kinematics and kinematics. Although the reviewer agrees that the participant can walk more naturally with a self-selected natural speed, speed still could be a confounding factor in this study since different participants might walk with significantly different speeds. Please discuss this point in the Discussion section.

* In the three sub-figures in the second column in Figure 2 and all sub-figures in Figure 4, what is the meaning of the white rectangular with a dash-dotted edge? Please explain in the figure legends.

* Please discuss in the Discussion section that how the changes in kinematics and kinetics with added masses found in this study can influence the design of exoskeletons in any aspects that you can think of, since this is the motivation of this study.
